# Trends in LVAD Placements and Outcomes: A Nationwide Analysis Using the National Inpatient Sample and National Readmissions Database

**DOI:** 10.3390/medsci13020060

**Published:** 2025-05-12

**Authors:** Vivek Joseph Varughese, Vignesh Krishnan Nagesh, Hadrian Hoang-Vu Tran, Nikita Wadhwani, Audrey Thu, Simcha Weissman, Adam Atoot

**Affiliations:** 1Department of Internal Medicine, University of South Carolina, Prisma Health, 2 Med Park, Richland, WA 29203, USA; vivekjvarughese@gmail.com; 2Department of Internal Medicine, Hackensack Palisades Medical Center, North Bergen, NJ 07047, USA; vgneshkrishnan@gmail.com (V.K.N.); nikita.wadhwani@hmhn.org (N.W.); simcha.weissman@hmhn.org (S.W.); adam.atoot@hmhn.org (A.A.); 3Touro College of Osteopathic Medicine, New York, NY 10027, USA; sphyo@student.touro.edu

**Keywords:** LVAD, ECMO, heart failure

## Abstract

Background: Aim of the study is to analyze the trends and outcomes in Left Ventricular Assist Device (LVAD) placements between the years 2016 and 2022 using the National Inpatient Sample (NIS). Methods: Using the NIS for the years 2016–2022, we identified the total number of LVAD placements using the PCS 10 code 02HA0QZ. In-hospital outcomes and healthcare resource utilization burden were assessed. Stratification of outcomes with Extracorporeal Membrane Oxygenation (ECMO) support were performed for the years 2018–2022. Outcome analysis variance in admissions requiring ECMO support was performed using multivariate regression analysis. A two tailed *p*-value < 0.05 was used to determine statistical significance. Results: A general decreasing trend was observed in the total number of LVAD placements, with 852 total admissions requiring LVAD placements identified in 2016 compared to 665 in 2022. The admissions for LVAD placements requiring ECMO support had an increasing trend, with 2.21% of admissions needing ECMO support in 2018 compared to 12.18% in 2018. After multivariate regression analysis, the association between all-cause mortality during the hospital stay for LVAD placements and requirement of ECMO was found to be significant, with an odds ratio of 2.34 (1.83–4.42, *p*-value: 0.001). Conclusions: A general decreasing trend in LVAD placements was observed between 2016 and 2022. All-cause mortality and hospital charges during the admission had a stable trend over the years. The requirement of ECMO support had an increasing trend from 2018 to 2022. Requirement of ECMO support during the admissions for LVAD placements had a statistically significant association with all-cause mortality during the admission. A 11.50% readmission rate was observed in the 30 days following discharge, with heart failure being the major cause of readmission.

## 1. Introduction

Over the last several decades, Left Ventricular Assist Device (LVAD) technologies have progressed from a conceptual theory in failed cardiopulmonary bypass to an accepted therapy for advanced heart failure patients. Accelerated use of continuous-flow LVADs (CF-LVADs) for long-term support has changed the face of advanced heart failure care [1]. The year 1953 marked the beginning of a modern era in cardiac surgery, when a cardiopulmonary bypass was introduced that allowed complex operations to be carried out. The concept of cardiopulmonary bypass was crucial and worked as the foundational principle for circulatory assist devices. By the 1960s, simple cardiac assist devices began to replace cardiopulmonary bypass in the management of post-cardiotomy shock. Liotta et al. reported the first clinical use of implantable artificial ventricle. This primitive ventricular assist device (VAD) consisted of a pneumatically driven, tubular displacement pump with a valved conduit connecting the left atrium to the descending aorta [2]. By 1966, the first successful pneumatic LVAD had been used to support a patient for 10 days after complex cardiac surgery [3]. Although LVAD technology was pioneered in the bridge setting where transplant offered a bailout for device failure, as devices developed, development began to be targeted towards devices capable of long-term or permanent circulatory support [4]. Over the past three decades, a series of revolutions in pump design and pivotal clinical trials have changed the face of advanced heart disease care. The landmark FDA approval of the HM XVE for permanent destination therapy (DT) in 2003 uncoupled access to durable Mechanical Circulatory Support (MCS) from transplant eligibility. As VAD therapy entered the mainstream, the collaborative Interagency Registry of Mechanically Assisted Circulatory Support (INTERMACS) was established in 2006 to map the evolution of durable MCS [5]. Currently, there are three recognized indications for the use of Left Ventricular Assist Devices (LVADs): (1) bridge to transplantation (BTT); (2) destination therapy (DT) for patients not considered eligible for heart transplant; and (3) bridge to myocardial recovery allowing reverse remodeling [6]. The evolution of continuous-flow devices, particularly the HeartMate 3 (HM3), has dramatically improved survival in advanced heart failure patients. This has been reflected as a general increasing trend towards LVAD placements globally [7]. Magnetically levitated circulatory systems tend to perform better than the non-magnetic systems, and the HM3 devices have shown superiority in terms of hemocompatibility-related adverse events like thrombosis [8]. There are also predictive models, like the HM3 risk scores, and advanced imaging techniques have optimized patient selection criteria for LVADs [9]. Despite these advancements, line infections, bleeding complications, and right ventricular failure remain as challenges related to long-term management of patients with LVAD [10].

National Inpatient Sample (NIS) and the National Readmissions Database (NRD) are two of the largest publicly available databases in the United States (US). The aim of our study is utilizing the NIS databases for analyzing the trends in LVAD placements between 2016 and 2022, and analyzing outcomes by studying 30-day readmission events after LVAD placements using the NRD. The Extracorporeal Life Support Organization (ELSO), which is the largest worldwide database regarding ECMO outcomes, generally includes data from accredited centers of excellence. However, the use of ECMO in non-accredited and non-academic centers is becoming more prevalent. Analyzing the trends using the NIS gives an advantage in this respect. Also, Veno-arterial Extracorporeal Membrane Oxygenation (VA ECMO) has been used more for LVAD placements, and our study specifically looks at difference in patient outcomes for LVAD placements with associated VA ECMO used as bridging.

## 2. Methods

NIS for the years 2016–2022 and the NRD for the year 2021 were used for analysis. ICD (International Classification of Diseases) 10 PCS, procedure code 02HA0QZ, [11] was used to select admissions that underwent LVAD placement. NIS was used to stratify patients to study the population demographics. ICD 10 PCS, codes 5A1522G and 5A1522H, were used to identify admissions that needed Extracorporeal Membrane Oxygenation (ECMO) in addition to the LVAD placement. General in-hospital mortality for admissions that needed LVAD placements as well specific mortality with concomitant need for ECMO were analyzed. In the analysis of ECMO requirement in addition to LVAD, NIS databases for the years 2018–2022 were used due to changes in the ICD 10 PCS code for ECMO. The association between the use of ECMO and all-cause mortality during the admission was studied using multivariate logistic regression, after accounting for age, sex, race, and mortality predictor indices during admission. NRD was used to identify admissions undergoing LVAD procedures in 2021, and readmission events in the 30 days following discharge were stratified. Among the readmissions in the 30 days following LVAD placement, the ones for Heart Failure (HF) and related conditions were selected using ICD 10 codes I110, I120, I320, I5230 [11]. Index admissions for LVAD placements were stratified based on the documented readmissions for HF in the 30 days following discharge, and prevalence of comorbid conditions were analyzed in both the groups. One-way ANOVA was used to stratify the differences in the prevalence of comorbidities in both the groups (Kolmogorov–Smirnov test). A two tailed *p*-value < 0.05 was used to determine statistical significance of difference in comorbidities. Comorbidities that held significant difference between the groups were analyzed using logistic regression, analyzing the association with HF readmissions in the 30 days following discharge. Multivariate regression analysis (probit) was used for comorbidities that held significant association with HF readmissions, after accounting for age, sex, race, monthly income, and comorbidities that held significant differences in the ANOVA. *p*-value < 0.05 was used to determine statistical significance.

## 3. Results

The general trends and demographic analysis of LVAD placements are summarized in Table 1. A general declining trend was observed in the total number of LVAD placements, with 665 total admissions requiring LVAD placements in 2022 compared to 852 in 2016. Meanwhile, an increasing trend was observed in the utilization of VA ECMO associated with LVAD placements: 2.21% of LVAD placements in 2018 required VA ECMO support compared to 12.18% in 2018. All-cause mortality, hospital length of stay, as well hospital charges remained statistically similar between 2016 and 2022.

Analyzing the national trends, there has been a declining trend in the total number of LVAD placements between 2016 and 2022, with 894 total documented LVAD placements in 2017, and 852 LVAD placements in 2016, compared to 665 in 2022 (Figure 1).

The trends for all-cause mortality during the admissions that required LVAD placements between 2016 and 2022 are depicted in Figure 2. The general all-cause mortality held no statistically significant difference across the years 2016–2022. One-way ANOVA for all-cause mortality across the years proved no variance over the years (*p*-value 0.872).

An upward trend was observed in the length of hospital stay (in days) as well as the total hospital charges associated with the admissions requiring LVAD placements between 2016 and 2022 (Figure 3).

A general upward trend in the utilization of Extracorporeal Membrane Oxygenation (ECMO) in addition to LVAD support was observed between 2018 and 2022 (Figure 4). Around 12.8% (9.90–14.8), i.e., around 80 admissions, required ECMO support in addition to LVAD in 2022, compared to 2.21% (1.58–4.32) in 2018.

The all-cause in-hospital mortality for admissions requiring ECMO support in addition to LVAD was significantly higher, and the trend remained consistent between 2018 and 2022 (Figure 5). Multivariate regression models were used to analyze the association between ECMO support and all-cause mortality during the admission for LVAD placements. Age, sex, race, and mortality risk predictive indices during admission (APR DRG mortality index) were used in the regression analysis. After accounting for the above-mentioned confounders, requirement of ECMO support was found to have a significant association with all-cause mortality in LVAD admissions, with an adjusted odds ratio of 2.34 (95% confidence interval: 1.83–4.42, *p*-value: 0.001).

In all, 1783 index admissions requiring LVAD support during the admission were identified in the 2021 NRD, and 30-day readmission events were analyzed following discharge. A total of 205 readmission events were documented (11.50%). The major readmission causes in the 30 days following discharge are depicted in Figure 6. Heart Failure was the major cause of 30-day readmission, followed by bleeding, transplant rejection, cardiogenic shock, and sepsis. Other readmission events noted as per the NRD were Acute Kidney Injury (AKI), COVID, and pneumonia.

Multivariate regression analysis was performed to analyze patient comorbidities associated with 30-day readmissions for HF. No patient factors were found to hold a significant association after the regression analysis.

## 4. Discussion

A general declining trend in the total number of LVAD placements was observed between 2016 and 2022 (Table 1, Figure 1). This has been consistent with the findings of the Society of Thoracic Surgeons, Interagency Registry for Mechanically Assisted Circulatory Support (INTERMACS) report for the years 2012 to 2022 [12]. As per the report, there was a 23.5% decrease in the volume of LVAD placements in 2022 compared to 2021. The change is attributed to the COVID pandemic and the changes in the US heart transplant allocation system. However, looking at Figure 5, although there was a decreasing trend in the total number of LVAD placements, likely related to the COVID pandemic, it was not reflected in the all-cause in-hospital mortality related with LVAD placements. The last several years have been characterized by a shift in device indication and type, with 81.1% of patients now implanted as destination therapy and 92.7% receiving an LVAD with full magnetic levitation. A similar study to ours was carried out utilizing the NIS for the years 2008 to 2016 [13]. As per the results of this large database analysis, there was a threefold increase in the total number of LVAD placements in 2016 compared to 2008. In-hospital mortality in patients with left ventricular assist devices decreased from 19.6% in 2008 to 8.1% in 2016 and was found to be higher at low-volume institutions compared with high-volume institutions. A general upward trend in the length of hospital stay and total hospital charges was observed in admissions requiring LVAD support (Figure 3). The mean length of hospital stay was 37.15 in 2016 compared to 45 days in 2022, but the mean duration of stay did not differ between the years in the variance analysis (one-way ANOVA).

A general upward trend was observed in the requirement of ECMO support during LVAD admissions between the years 2018 and 2022 (Figure 4). A study using the Extracorporeal Life Support Organization (ELSO) registry data from 2010 to 2019 demonstrated that the overall use of ECMO increased from 1.7% in 2010 to 22.2% in 2019 among patients who were bridged to LVAD or orthotopic heart transplant (OHT) [14]. This increase highlights the growing reliance on ECMO for temporary circulatory support in critically ill patients. The 2018 United Network for Organ Sharing (UNOS) heart transplant allocation algorithm, which prioritizes Veno-arterial ECMO (VA-ECMO) patients, has also influenced these trends. After 2018, there has been an increase in the proportion of patients receiving ECMO as a bridge to LVAD or transplant, with a noted decrease in clinical acuity among these patients [15]. Analyzing the trends in all-cause mortality in LVAD patients requiring ECMO support, a general upgoing trend was observed. In another INTERMACS study analyzing the trends of mortality in LVAD admissions requiring ECMO support [16], a higher proportion of mortality was observed related to ECMO support. This trend can be attributed to the higher acuity of illness in the ECMO group. Extracorporeal membrane oxygenation (ECMO) as a bridge to left ventricular assist device (LVAD) implantation has shown promise in improving end-organ function and optimizing outcomes in some critically ill patients, but the practice remains controversial. As per our analysis, LVAD admissions requiring ECMO support were found to have a significant association with in-hospital mortality, after regressing for confounders. However, ECMO support was found to improve the hemodynamics significantly in patients needing LVAD. In another retrospective analysis performed using the INTERMACS registry in 2018, LVAD patients requiring ECMO support were found to have improved central pulmonary pressures and mean pulmonary artery pressures [17]. However, a limitation of our study was that the temporal association of ECMO support to the need of LVAD could not be determined due to the nature of the database.

Analyzing 30-day readmissions following the LVAD placement, a 11.5% readmission rate was observed. HF was the major cause of readmission. Bleeding, transplant rejection, cardiogenic shock, sepsis, etc., were the other major readmission causes. Readmission events for acute kidney injury, community acquired pneumonia, COVID were the other causes of 30-day readmissions. A similar study using the NRD for the years 2013 and 2014 was conducted, in which a 53.9% rate of readmission was observed in the 90 days following discharge [18]. A longer length of hospital stay and discharge to acute care nursing facilities were found to be factors that held significant association with readmissions. However, our analysis did not document any association between preprocedural patient comorbidities and 30-day readmissions. Procedure-related bleeding, transplant rejection, cardiogenic shock, and sepsis were the other causes of readmission in the 30 days following discharge. Analyzing other retrospective studies, bleeding, volume overload, and transplant rejection were found to be the major readmission causes in the one year following LVAD placements. A study on HeartMate 3 LVAD patients found that 92% of patients experienced unplanned readmissions, with a mean cumulative number of 0.43 readmissions at one year and 1.13 readmissions at 900 days. The most frequent causes were major infections (29.3%) and bleeding (13.2%) [19]. In the MOMENTUM 3 Trial [20] studying Heart Mate III devices, a lower adverse event burden was observed compared to previous trials on LVADs.

Limitations of the Study: While the utility of NIS and NRD in large sample population studies are high, the databases are not without limitations. The patient selection criteria are based on ICD 10 and PCS codes, and there is a potential chance of bias associated with the inter-operator variability in coding discharge diagnoses. However, since major procedure codes like LVAD placements are less likely to undergo inter-provider bias, this is less likely to affect the outcomes of our analysis. However, the type of LVAD or generation of the device is not mentioned in the NIS database, preventing further stratification and analysis. Other limitations include the fact that trends were observed only for hospitals in the United States (US) and were limited to the years 2016 to 2022.

## 5. Conclusions

A general declining trend in the total number of LVAD placements was observed between 2016 and 2022—a 21.9% decrease in 2022 compared to 2016. All-cause mortality during admission remained similar over the years between 9 and 12%. A higher proportion of mortality was seen in LVAD patients requiring ECMO support. The requirement of ECMO support was found to have a significant association with all-cause mortality among LVAD admissions (OR: 2.34, 1.83–4.42). A 11.50% 30-day readmission rate was observed among LVAD admissions. Heart Failure remained the major cause of 30-day readmissions post-LVAD placements; however, no patient factors held significant association with heart failure readmissions in the multivariate regression analysis.

## Figures and Tables

**Figure 1 medsci-13-00060-f001:**
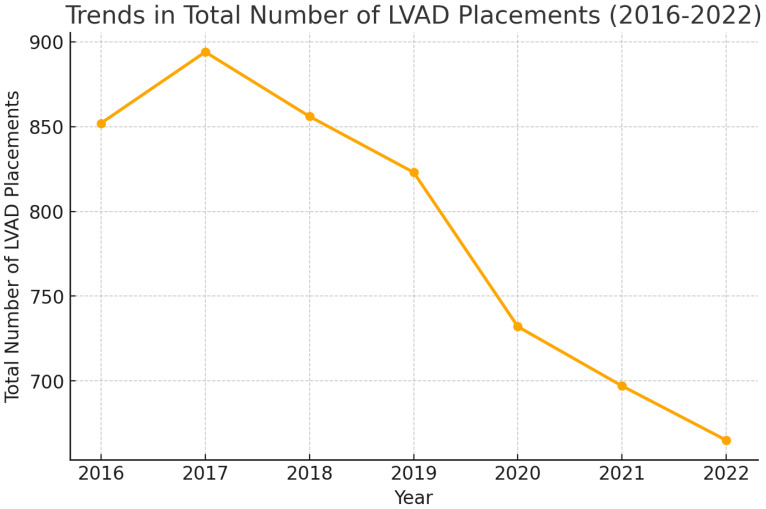
National Trends in LVAD (Left Ventricular Assist Device) placements (2016–2022).

**Figure 2 medsci-13-00060-f002:**
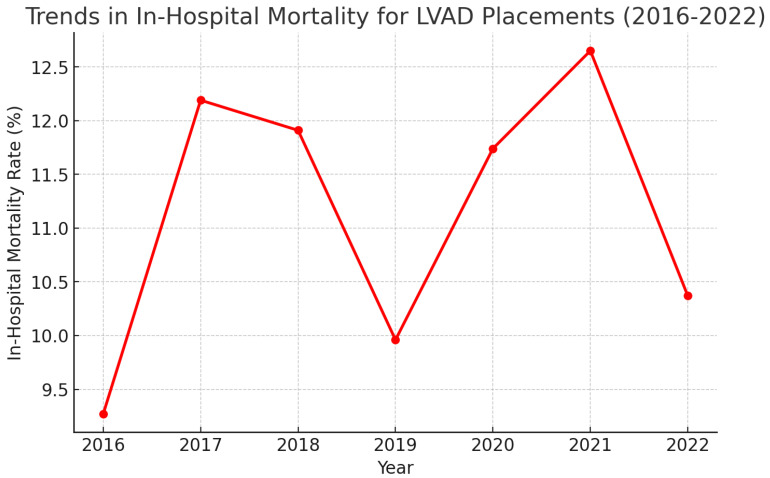
Trends in all-cause in-hospital mortality following LVAD placements (2016–2022).

**Figure 3 medsci-13-00060-f003:**
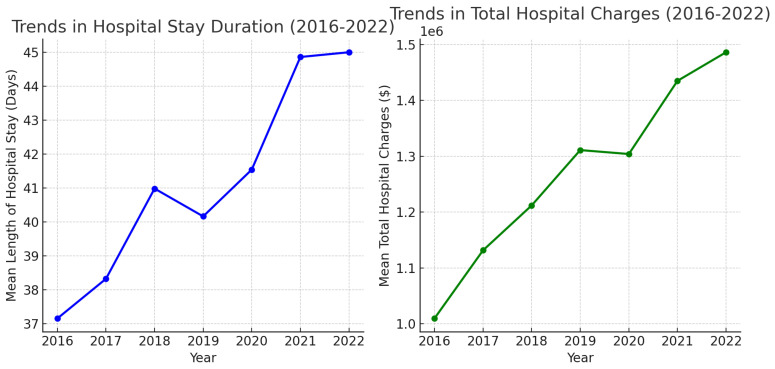
Trends in in-hospital length of stay and total hospital charges (2016–2022).

**Figure 4 medsci-13-00060-f004:**
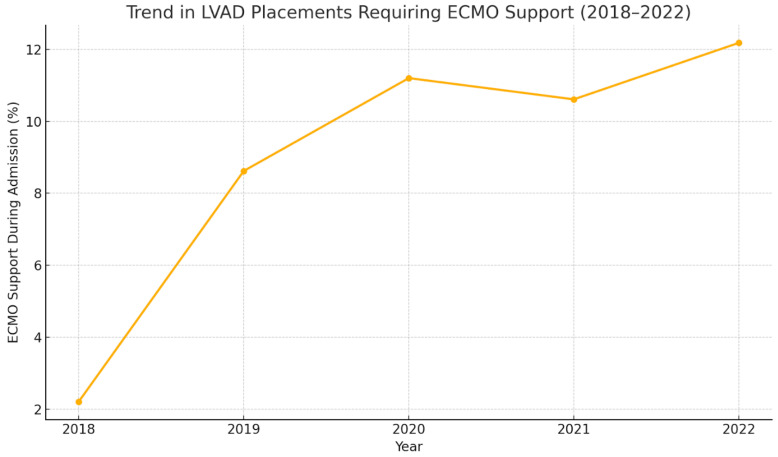
Trends in the use of Extracorporeal Membrane Oxygenation (ECMO), 2018–2022.

**Figure 5 medsci-13-00060-f005:**
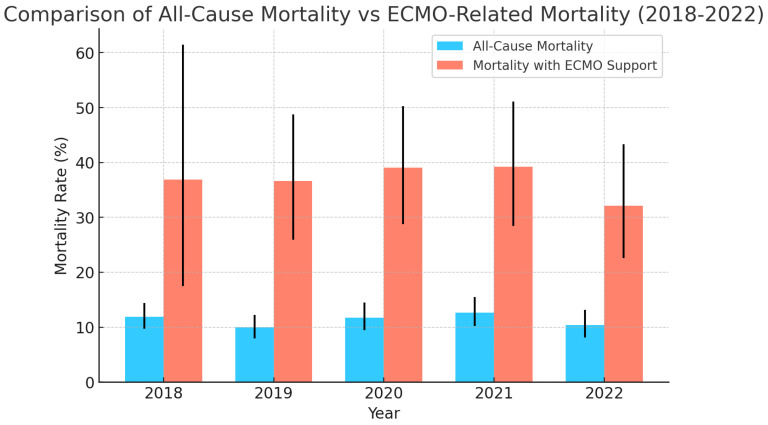
All-cause mortality in admissions requiring ECMO (Extracorporeal Membrane Oxygenation) support.

**Figure 6 medsci-13-00060-f006:**
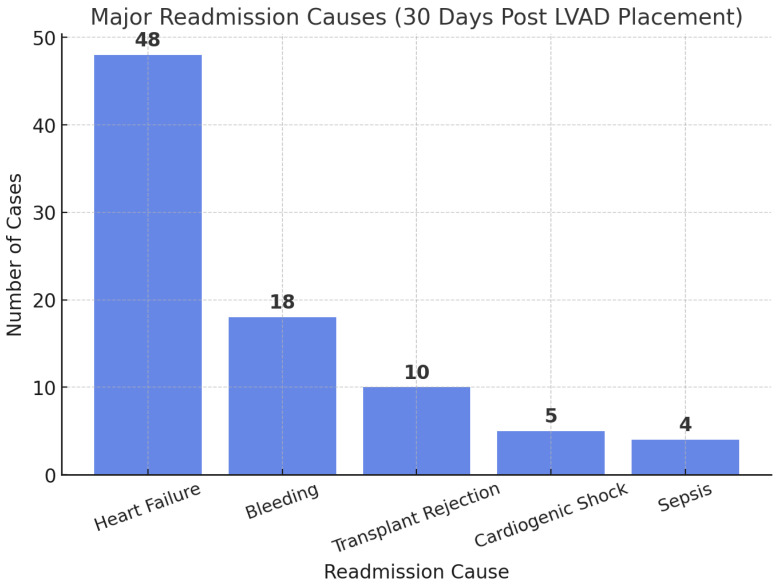
Thirty-day readmission events following the LVAD placement.

**Table 1 medsci-13-00060-t001:** Trends in LVAD placement (2016–2022) (LVAD: Left Ventricular Assist Device, ECMO: Extracorporeal Membrane Oxygenation).

Year	Total Number of LVAD Placements	Mean Age	Mean Length of Hospital Stay (Days)	Mean of Total Hospital Charges ($)	In-Hospital Mortality (All-Cause) %	Elective Procedures %	LVAD Placement Requiring ECMO Support During Admission %	In-Hospital Mortality for LVAD Placements Requiring ECMO Support %
2016	852	55.45	37.15	1,009,442	9.27% (7.49–11.41%)	25.55% (22.53–28.40%)	-	-
2017	894	55.31	38.32	1,131,464	12.19% (10.20–14.50%)	28.76% (25.88–31.83%)	-	-
2018	856	54.84	40.98	1,211,810	11.91% (9.90–14.26%)	24.12% (21.36–27.11%)	2.21% (1.58–4.32%)	36.84% (17.67–61.30%)
2019	823	55.34	40.16	131,092	9.96% (8.09–12.08%)	25.67% (22.43–28.78%)	8.62% (7.93–12.47%)	36.61% (26.11–48.56%)
2020	732	54.1	41.54	1,303,891	11.74% (9.60–14.29%)	25.03% (22.02–28.30%)	11.20% (9.45–14.53%)	39.02% (28.97–50.10%)
2021	697	55.7	44.86	1,434,880	12.65% (10.35–15.30%)	26.54% (23.39–29.95%)	10.61% (9.84–15.67%)	39.18% (28.61–50.88%)
2022	665	54.67	45.0	1,486,108	10.37% (8.27–12.93%)	24.73% (21.59–28.17%)	12.18% (9.90–14.89%)	32.09% (22.74–43.15%)

## Data Availability

No new data were created or analyzed in this study.

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
