# Peer review of "Trends in LVAD Placements and Outcomes: A Nationwide Analysis Using the National Inpatient Sample and National Readmissions Database"

_medsci, 2025, doi:10.3390/medsci13020060_

Round 1

Reviewer 1 Report

Comments and Suggestions for Authors

The current manuscript assessed the trends and consequences of the use of LVAD usage in one country from 2016 to 2022 retrospectively. A general decrease in total LVAD placements was reported, while all-cause mortality during admission remained almost constant. Moreover, LVAD patients requiring ECMO support showed higher mortality rates. The paper contains several issues that require attention.

  1. The introduction should include more relevant research and the novelty should be stated clearly.
  2. It is not clear what types of LVADs were used for the patients of the current study. All the results are dependent on the type of LVADs. For better judgment, it is recommended that more details on the number of each LVAD type be provided. Such data can be shown in a separate table or figure.
  3. The abstract is not included in the manuscript.
  4. Line 1: The type of the paper should be correct. This is not a “review” paper and seems to be original research on the usage of LVAD.
  5. Line 11: Why “Left Ventricular Assist Device” are written in capital letters while all other abbreviations are not so? Also, in the abstract, capital letters are used.
  6. Line 13-14: “(LVADs)” should be replaced by “(CF-LVADs)”
  7. Line 15: “.[1]” should be “[1].”
  8. Lines 15 and 16: “1953 marked a modern era in cardiac surgery, when cardiopulmonary bypass was introduced that allowed complex operations to be carried out.” This sentence should be supported by a reference.
  9. Lines 20 and 21: “ventricle assist device (VAD)” should be “ventricular assist device (VAD)”.
  10. Line 30: Did the authors mean “mechanical circulatory support (MCS)” instead of “mechanical circulatory systems (MCS)”?
  11. Lines 28-33: These two claims about DT in 2003 and INTERMACS in 2006 should be supported by relevant references.
  12. Line 36: “(LVAD)” should be replaced by “(LVADs)”
  13. The authors are recommended to provide a reference and double-check the international classification of diseases (ICD) codes (Lines 45-59). Also, in Line 46: did the authors mean “ICD-10-PCS Procedure Code 02HA0QZ” instead of “10 procedure code O2HA0QZ”?
  14. Line 59: “ICS 10 codes — --.” Needs to be completed, and as mentioned before, with supporting references.
  15. Line 74: Table 1 caption should be corrected and abbreviations should be presented correctly (ECM in the caption should be corrected to ECMO). Additionally, the third column heading, "Mean Age," is aligned closely with its adjacent columns without sufficient spacing, making it difficult to distinguish between the headings of columns 2, 3, and 4.
  16. In Figure 4, the horizontal axis shows years. Therefore, decimals should be removed (For example, year 2018.5 does not make sense!)
  17. Line 112: the authors should double-check the percentage reported here. 205*100/1783=11.50% not 11.64%. Be aware that this number is also mentioned in the abstract.
  18. In Figure 6, a total number of 85 (48+18+10+5+4=85) cases with major complications are reported. However, it is stated (in Line 112) that “A total of 205 readmission events were documented.”. The authors are recommended to at least name other causes of readmissions that are not shown in this figure.
  19. Are the all-cause mortality rates (2018-2022) in Figure 5 exactly the same as those provided in Figure 2 (2016-2022)? Surprisingly, it seems that the outbreak of COVID-19 did not change the mortality rates significantly! Although the impact of the COVID pandemic on LVAD placements is discussed in Line 125, the effect of COVID-19 should be added to the discussions for Figures 2 and 5.
  20. Discussions in Lines 139-141 seem to be incorrect and contradictory.
  21. Line 149: The authors should describe the abbreviation VA-ECMO, which is introduced here for the first time. Venoarterial extracorporeal membrane oxygenation (VA-ECMO)
  22. More limitations of this study should be added to the discussions section. For example: this study was only done in one country - only from 2016 to 2022 – only 30-day readmissions were analyzed, …
  23. English errors should be corrected. For example, Line 98: “all cause”, Line 103: “was used” >> “were used”, Line 103: “above mentioned” >> “above-mentioned”, Line 110: rewrite the sentence to make it clear and correct., Line 111 and Figure 6 caption: “30 day” >> “30-day”, Line 130: “three fold” >> “three-fold”, Line 153-155: “Another INTERMACS study … was observed.”, …
  24. The authors should pay attention to the correct usage of spaces between the characters throughout the manuscript. There are both redundant spaces (for example, Line 84, 135: “all- cause”, Line 89: “( Figure 3)”, Line 93: “( Figure 4)”, Line 95: “2.21 %”, Line 122: “( Table 1, Figure 1)”, Line 139: “( Figure 3)”, …) and dropped spaces (for example, Line 82: “Figure 2.The general”, Line 129: “[8].As per”, …).
  25. Ethics approval and funding details are not stated.
Comments on the Quality of English Language
  1. English errors should be corrected. For example, Line 98: “all cause”, Line 103: “was used” >> “were used”, Line 103: “above mentioned” >> “above-mentioned”, Line 110: rewrite the sentence to make it clear and correct., Line 111 and Figure 6 caption: “30 day” >> “30-day”, Line 130: “three fold” >> “three-fold”, Line 153-155: “Another INTERMACS study … was observed.”, …
  2. The authors should pay attention to the correct usage of spaces between the characters throughout the manuscript. There are both redundant spaces (for example, Line 84, 135: “all- cause”, Line 89: “( Figure 3)”, Line 93: “( Figure 4)”, Line 95: “2.21 %”, Line 122: “( Table 1, Figure 1)”, Line 139: “( Figure 3)”, …) and dropped spaces (for example, Line 82: “Figure 2.The general”, Line 129: “[8].As per”, …).

Author Response

  1. The introduction should include more relevant research and the novelty should be stated clearly.

Reply: changes made 

  1. It is not clear what types of LVADs were used for the patients of the current study. All the results are dependent on the type of LVADs. For better judgment, it is recommended that more details on the number of each LVAD type be provided. Such data can be shown in a separate table or figure.

Reply:  The nature of the NIS does not let us stratify based on the type of LVAD. This has been included as a limitation in the limitations section 

  1. The abstract is not included in the manuscript.
  2. Reply:  has been added
  1. Line 1: The type of the paper should be correct. This is not a “review” paper and seems to be original research on the usage of LVAD.

Reply: Changes have been made 

  1. Lines 15 and 16: “1953 marked a modern era in cardiac surgery, when cardiopulmonary bypass was introduced that allowed complex operations to be carried out.” This sentence should be supported by a reference.

Reply:  this was mentioned in the study Liotta et al [2]

  1. Line 30: Did the authors mean “mechanical circulatory support (MCS)” instead of “mechanical circulatory systems (MCS)”?

reply:  changes have been made 

  1. Line 36: “(LVAD)” should be replaced by “(LVADs)”
  2. Reply:  changed 
  1. The authors are recommended to provide a reference and double-check the international classification of diseases (ICD) codes (Lines 45-59). Also, in Line 46: did the authors mean “ICD-10-PCS Procedure Code 02HA0QZ” instead of “10 procedure code O2HA0QZ”?

  2. Reply: 
    changes have been made 
  1. Line 59: “ICS 10 codes — --.” Needs to be completed, and as mentioned before, with supporting references.

  2. Reply: 
    changes have been made 
  1. Line 74: Table 1 caption should be corrected and abbreviations should be presented correctly (ECM in the caption should be corrected to ECMO). Additionally, the third column heading, "Mean Age," is aligned closely with its adjacent columns without sufficient spacing, making it difficult to distinguish between the headings of columns 2, 3, and 4.

  2. Reply: 
    changes have been made 
  1. Line 112: the authors should double-check the percentage reported here. 205*100/1783=11.50% not 11.64%. Be aware that this number is also mentioned in the abstract.

  2. Reply: 
    changes have been made 
  1. In Figure 6, a total number of 85 (48+18+10+5+4=85) cases with major complications are reported. However, it is stated (in Line 112) that “A total of 205 readmission events were documented.”. The authors are recommended to at least name other causes of readmissions that are not shown in this figure.

  2. REPLY: 
    The other readmission causes we obtained using the NRD were isolated cases ( having only 1-2 cases documented, ie, pneumonia, AKI etc) hence did not include 

  1. Are the all-cause mortality rates (2018-2022) in Figure 5 exactly the same as those provided in Figure 2 (2016-2022)? Surprisingly, it seems that the outbreak of COVID-19 did not change the mortality rates significantly! Although the impact of the COVID pandemic on LVAD placements is discussed in Line 125, the effect of COVID-19 should be added to the discussions for Figures 2 and 5.

  2. Reply: 
    Yes, the all cause mortality in all LVAD placements are similar in figure 2 and 5. ECMO related mortality is higher in figure 5 which was not mentioned in figure 2. Ex: all cause mortality for 2018 in figure 2 was 11%, which is seen in Figure 5 also. The discussion part mentioned impact on total number of LVAD placements during COVID, however, the pandemic did not materialize as increase in mortality rates, likely because of the lower number of LVAD attempts done during pandemic 

  1. Line 149: The authors should describe the abbreviation VA-ECMO, which is introduced here for the first time. Venoarterial extracorporeal membrane oxygenation (VA-ECMO)

  2. Reply: 
    changes have been made 
  1. More limitations of this study should be added to the discussions section. For example: this study was only done in one country - only from 2016 to 2022 – only 30-day readmissions were analyzed, …

Reply: added 

  1. English errors should be corrected. For example, Line 98: “all cause”, Line 103: “was used” >> “were used”, Line 103: “above mentioned” >> “above-mentioned”, Line 110: rewrite the sentence to make it clear and correct., Line 111 and Figure 6 caption: “30 day” >> “30-day”, Line 130: “three fold” >> “three-fold”, Line 153-155: “Another INTERMACS study … was observed.”, …
  2. The authors should pay attention to the correct usage of spaces between the characters throughout the manuscript. There are both redundant spaces (for example, Line 84, 135: “all- cause”, Line 89: “( Figure 3)”, Line 93: “( Figure 4)”, Line 95: “2.21 %”, Line 122: “( Table 1, Figure 1)”, Line 139: “( Figure 3)”, …) and dropped spaces (for example, Line 82: “Figure 2.The general”, Line 129: “[8].As per”, …).

REPLY: changes have been made 

  1. Ethics approval and funding details are not stated.

REPLY:  will add 

Reviewer 2 Report

Comments and Suggestions for Authors

The analysis shows a 30-day readmission rate of 11.64%, with heart failure being a primary cause of readmission. The declining trend in LVAD placements raises concerns about access to and utilization of advanced heart failure treatments, potentially worsened by factors such as the COVID-19 pandemic and changes in heart transplant allocation systems. The stability in in-hospital mortality indicates that while fewer patients receive LVADs, those who do are not experiencing worse outcomes than in previous years.

Can the stability of in-hospital mortality be attributed to increased mortality at home during the COVID-19 pandemic?  (resulting in many patients not reaching the hospital)

However, the increased mortality linked to ECMO support underscores the complexities of managing critically ill patients and the necessity for improved preoperative risk assessment.

The article highlights the significant association between the need for ECMO support and increased mortality rates. The authors conclude, "LVAD placements requiring ECMO support were found to have a significantly higher mortality." The study indicates that patients requiring ECMO support often present with greater illness severity, contributing to their increased risk of mortality. Specifically, ECMO is typically used in more critically ill patients who may have more severe underlying conditions or complications.

 Could the increase in mortality with ECMO be due to complications generated by its use?

What underlying factors contribute to the decline in LVAD placements over the analyzed years?

How do patient demographics and comorbidities influence the outcomes of LVAD placements and associated mortality rates?

What interventions can be implemented to reduce the 30-day readmission rates for patients post-LVAD placement?

How does ECMO support impact all-cause mortality in LVAD patients?

What statistical methods were used to analyze the data from the NIS and NRD?

Author Response

Can the stability of in-hospital mortality be attributed to increased mortality at home during the COVID-19 pandemic?

REPLY:  Very much. And also attributed to the decreased LVAD attempts made during the pandemic time. This has been discussed in the discussion

 Could the increase in mortality with ECMO be due to complications generated by its use? 

REPLY:  potentially yes. But the Charleson comorbidity score (a cumulative score derived from patient comorbidities) and PADRG risk severity indices were used in the regression analysis to account for this. It could be related to complication related to the use as well the myocardial stunning during VA ECMO support/ This has been added in the discussion part 

What statistical methods were used to analyze the data from the NIS and NRD?

REPLY:  specified in the methods section

How do patient demographics and comorbidities influence the outcomes of LVAD placements and associated mortality rates?

REPLY:  patient race, sex, comorbidities ( APDRG risk index and Charleson Comorbidity indices), median quarterly income were used in the regression analysis)

What interventions can be implemented to reduce the 30-day readmission rates for patients post-LVAD placement?

REPLY:  this would be more appropriate in prospective or retrospective real-life studies due to the nature of the database and limitations of patient variables we had 

Round 2

Reviewer 1 Report

Comments and Suggestions for Authors

Despite making revisions, the revised version shows that the authors have ignored a number of comments or addressed some partially. The authors should not ignore previous comments without providing reasons and should address all comments either by answering directly to the comment or by revising the manuscript. Additionally, showing tracked changes makes the revised version hard to follow. The authors are recommended to highlight changes in the revised version instead of using the 'Track Changes' option in Microsoft Word.

  1. Line 81: the ICD PCS code does not seem to be correctly written. It is named “O2HA0QZ” while the abstract says “02HA0QZ”. There are still no references that readers can read more about these codes. Also, Line 81 shows “ICD PCS 10 code …”, while Line 83 shows “ICD 10 PCS code …”. What is the correct form in the original reference?
  2. The introduction should include more relevant research and the novelty should be stated clearly. You should explain what has been done in this research that has not been addressed in previous research.
  1. Still, sentences in lines 36-40 and 53-59 lack a reference in the revised manuscript. Also, do statements in Lines 69-72 come from ref [7]? Authors should show this in the revised version.
  2. In Figure 4, the horizontal axis shows years. Therefore, decimals should be removed (For example, the year 2018.5 does not make sense!)
  3. As mentioned before, Line 63 should be (LVADs) not (LVAD).
  4. Line 225: the incorrect percentage “11.64%” still exists.
  5. In the previous comment, the authors were asked to add some major complications that are not included in the figure to the revised version. They just answered why they did not include these! “In Figure 6, a total number of 85 (48+18+10+5+4=85) cases with major complications are reported. However, it is stated (in Line 112) that “A total of 205 readmission events were documented.”. The authors are recommended to at least name other causes of readmissions that are not shown in this figure.” The authors answered that REPLY: The other readmission causes we obtained using the NRD were isolated cases ( having only 1-2 cases documented, ie, pneumonia, AKI etc) hence did not include.
  6. The limitations mentioned in the previous comments have been ignored and not added to the revised version. The authors should have mentioned as many limitations as possible. “More limitations of this study should be added to the discussions section. For example this study was only done in one country - only from 2016 to 2022 – only 30-day readmissions were analyzed, … ”.
  7. Discussions in Lines 175-177 seem to be incorrect and contradictory.
  8. If the authors remove the INTERMACS in Line 56, they should define it at its first appearance in the revised version.
  9. “Limitations” should be included as a subsection within the “Discussion” section. It is also recommended that all limitations be expressed in this subsection. Therefore, limitations mentioned in Lines 201-203 should move to this subsection.
  10. Line 111 (table 1 caption): “;” should be replaced by “:”.
  11. English errors should be corrected. For example: “it” in Line 232, “In-hospital” in Line 85, “The all cause in hospital” in Line 134, …
  12. The space issue still persists in some places. For example: Line 171, Line 120, Line 225, Line 131, …
  13. There is a font and style issue in Lines 66-67.
  14. Some texts have different colors (gray). For example, Lines 179-192, 215-219.
Comments on the Quality of English Language

English errors should be corrected. For example: “it” in Line 232, “In-hospital” in Line 85, “The all cause in hospital” in Line 134, …

Author Response

  1. Sorry for the mistake. Think the wrong file was uploaded prior to this: all the changes that were mentioned in the previous review as well are the current review are marked in red.
  2.  
  3.  
  4. Line 81: the ICD PCS code does not seem to be correctly written. It is named “O2HA0QZ” while the abstract says “02HA0QZ”. There are still no references that readers can read more about these codes. Also, Line 81 shows “ICD PCS 10 code …”, while Line 83 shows “ICD 10 PCS code …”. What is the correct form in the original reference?REPLY:   changes made 

The introduction should include more relevant research and the novelty should be stated clearly. You should explain what has been done in this research that has not been addressed in previous research.

New references added to the intro section 

Still, sentences in lines 36-40 and 53-59 lack a reference in the revised manuscript. Also, do statements in Lines 69-72 come from ref [7]?

REPLY: Yes 

  1. In Figure 4, the horizontal axis shows years. Therefore, decimals should be removed (For example, the year 2018.5 does not make sense!)

Changes made 

  1. As mentioned before, Line 63 should be (LVADs) not (LVAD).
  2. Line 225: the incorrect percentage “11.64%” still exists.

REPLY:  changes made 

In the previous comment, the authors were asked to add some major complications that are not included in the figure to the revised version. They just answered why they did not include these! “In Figure 6, a total number of 85 (48+18+10+5+4=85) cases with major complications are reported. However, it is stated (in Line 112) that “A total of 205 readmission events were documented.”

REPLY:  changes added 

  1. The limitations mentioned in the previous comments have been ignored and not added to the revised version. The authors should have mentioned as many limitations as possible. “More limitations of this study should be added to the discussions section. For example this study was only done in one country - only from 2016 to 2022 – only 30-day readmissions were analyzed, … ”.

REPLY  : limitations section added 

  1. Discussions in Lines 175-177 seem to be incorrect and contradictory.
  2. If the authors remove the INTERMACS in Line 56, they should define it at its first appearance in the revised version.

REPLY: changes made 

  1. Line 111 (table 1 caption): “;” should be replaced by “:”.
  2. English errors should be corrected. For example: “it” in Line 232, “In-hospital” in Line 85, “The all cause in hospital” in Line 134, …
  3. The space issue still persists in some places. For example: Line 171, Line 120, Line 225, Line 131, …
  4. There is a font and style issue in Lines 66-67.
  5. Some texts have different colors (gray). For example, Lines 179-192, 215-219.

REPLY: issues corrected 

Round 3

Reviewer 1 Report

Comments and Suggestions for Authors

It seems that the authors are attempting to answer some of the reviewers' comments carelessly and ignore some others in each revision. They try to submit the revisions carelessly as soon as possible, but they forget to proofread the revised version and check for obvious errors and mistakes! This has made the paper even worse after being revised twice!
This time, there are even terrible issues with styles and fonts. Just please compare your v2 with v3 to see what happened to the styles and fonts of your manuscript!
The authors should reread all of my previous comments (first and second comments) and ensure that they have solved all the issues mentioned.

In this version, the abstract is again removed!
Table captions, references, author affiliations, ... have incorrect styles and fonts! Reference 11 is just a hyperlink and does not show more details! Affiliation numbers are not superscripts.
Figure 4, which is revised, is out of the page borders!
The limitations that I mentioned in the previous comments have not been added, and the limitations section is not a subsection of the discussion; instead, the authors have removed section numbers!
Why are all captions and manuscript title now written in capital letters?
Line 70: What does ** mean?
Line 74: Simirnov was corrected to Smirnov in v2, but now it is again incorrect in v3!
Line 57: O2HA0QZ or 02HA0QZ (zero or O letter)?
There are still some extra spaces in the text.

If the authors ignore the previously mentioned issues and submit a revision carelessly without proofreading next time, the paper will be rejected.

Author Response

Hi, 

Sorry for the mistake. I think there was some confusion regarding the version that was uploaded. This time, we have taken the time to individually go through all the suggestions. Hence, we are starting with the reviews made from round 1 to round 3 to make sure the changes are reflected. Reflected changes are made in red color: 

The introduction should include more relevant research, and the novelty should be stated clearly. Reply: 2 extra studies are quoted in the introduction, and an added paragraph on why NIS based studies would provide a different perspective than ELSO studies and association with ECMO mortality has been added. 

It is not clear what types of LVADs were used for the patients of the current study. All the results are dependent on the type of LVADs. For better judgment, it is recommended that more details on the number of each LVAD type be provided. Such data can be shown in a separate table or figure. Reply: Unfortunately, this is beyond the scope of the NIS/ NRD database: the PCS codes doesnt differentiate based on the type of LVAD, making it impossible for us to analyze based on this stratification. 

The abstract is not included in the manuscript. Reply: this was a technical mistake from our side. We got confused with the versions. Made sure that the current uploaded version has the abstract included 

Line 1: The type of the paper should be correct. This is not a “review” paper and seems to be original research on the usage of LVAD. Reply: Original research added on line 4 

Line 11: Why “Left Ventricular Assist Device” are written in capital letters while all other abbreviations are not so? Also, in the abstract, capital letters are used.  Reply: This was because LVAD was mentioned for the first time in the main text.

Line 13-14: “(LVADs)” should be replaced by “(CF-LVADs)” Reply:: changes made, marked in red 

Line 15: “.[1]” should be “[1].” Reply: change made, marked in red

Lines 15 and 16: “1953 marked a modern era in cardiac surgery, when cardiopulmonary bypass was introduced that allowed complex operations to be carried out.” This sentence should be supported by a reference.  Reply: The reference [2] includes this information/ ie was quoted from the same study 

Lines 20 and 21: “ventricle assist device (VAD)” should be “ventricular assist device (VAD)”. Reply: changes made and marked in red and bold. 

Line 30: Did the authors mean “mechanical circulatory support (MCS)” instead of “mechanical circulatory systems (MCS)”?  Reply: change made and marked in red 

Lines 28-33: These two claims about DT in 2003 and INTERMACS in 2006 should be supported by relevant references.  Reply: the Destination therapy and associated uncopuling with transplant was not a direct quote from study but more of a inference we had. About INTERMACS, reference in [5]

Line 36: “(LVAD)” should be replaced by “(LVADs)” Reply: changes made, marked in red 

The authors are recommended to provide a reference and double-check the international classification of diseases (ICD) codes (Lines 45-59). Also, in Line 46: did the authors mean “ICD-10-PCS Procedure Code 02HA0QZ” instead of “10 procedure code O2HA0QZ”?  Reply:  Refernce 11 is the link for the website. Changes made for ICD -10-PCS code and marked in red 

Line 59: “ICS 10 codes — --.” Needs to be completed, and as mentioned before, with supporting references.  Reply: chnages made, marked in red, reference is same as the ICD 10 official webiste 

Table 1 caption should be corrected and abbreviations should be presented correctly (ECM in the caption should be corrected to ECMO). Additionally, the third column heading, "Mean Age," is aligned closely with its adjacent columns without sufficient spacing, making it difficult to distinguish between the headings of columns 2, 3, and 4: Reply: changes made, marked in red 

In Figure 4, the horizontal axis shows years. Therefore, decimals should be removed (For example, year 2018.5 does not make sense!):  Reply: change made 

Line 112: the authors should double-check the percentage reported here. 205*100/1783=11.50% not 11.64%. Be aware that this number is also mentioned in the abstract. Reply: changes made and marked in red 

In Figure 6, a total number of 85 (48+18+10+5+4=85) cases with major complications are reported. However, it is stated (in Line 112) that “A total of 205 readmission events were documented.”. The authors are recommended to at least name other causes of readmissions that are not shown in this figure.  Reply: added. in red color 

Are the all-cause mortality rates (2018-2022) in Figure 5 exactly the same as those provided in Figure 2 (2016-2022)? Surprisingly, it seems that the outbreak of COVID-19 did not change the mortality rates significantly! Although the impact of the COVID pandemic on LVAD placements is discussed in Line 125, the effect of COVID-19 should be added to the discussions for Figures 2 and 5.  Reply: Yes: the all cause mortality in the range of 12-15 % in Figure 5 correlated with all cause mortality defined in table 1:  A extra point in the discussion section on how covid did not reflect a change in all cause mortality has been added ( marked in red) 

Discussions in Lines 139-141 seem to be incorrect and contradictory. Reply: removed 

Line 149: The authors should describe the abbreviation VA-ECMO, which is introduced here for the first time. Venoarterial extracorporeal membrane oxygenation (VA-ECMO)  Reply: added in red 

More limitations of this study should be added to the discussions section. For example: this study was only done in one country - only from 2016 to 2022 – only 30-day readmissions were analyzed, Reply:  Added, marked in bold red in limitations section. 

English errors should be corrected. For example, Line 98: “all cause”, Line 103: “was used” >> “were used”, Line 103: “above mentioned” >> “above-mentioned”, Line 110: rewrite the sentence to make it clear and correct., Line 111 and Figure 6 caption: “30 day” >> “30-day”, Line 130: “three fold” >> “three-fold”, Line 153-155: “Another INTERMACS study … was observed.”, …

The authors should pay attention to the correct usage of spaces between the characters throughout the manuscript. There are both redundant spaces (for example, Line 84, 135: “all- cause”, Line 89: “( Figure 3)”, Line 93: “( Figure 4)”, Line 95: “2.21 %”, Line 122: “( Table 1, Figure 1)”, Line 139: “( Figure 3)”, …) and dropped spaces (for example, Line 82: “Figure 2.The general”, Line 129: “[8].As per”, …).  : Reply: changes have been made 

Ethics approval and funding details are not stated. Reply: changes made, marked in red 

Line 81: the ICD PCS code does not seem to be correctly written. It is named “O2HA0QZ” while the abstract says “02HA0QZ”. There are still no references that readers can read more about these codes. Also, Line 81 shows “ICD PCS 10 code …”, while Line 83 shows “ICD 10 PCS code …”. What is the correct form in the original reference? Reply: ICD  10 PCS code made unform across main text and abstract: reference in [11]. 

The introduction should include more relevant research and the novelty should be stated clearly. You should explain what has been done in this research that has not been addressed in previous research. Reply: added 

If the authors remove the INTERMACS in Line 56, they should define it at its first appearance in the revised version. Reply: changed and marked in red 

Still, sentences in lines 36-40 and 53-59 lack a reference in the revised manuscript. Also, do statements in Lines 69-72 come from ref [7]? Authors should show this in the revised version.  Reply: We did not understand what this exactly means: we have mentioned ideas we deciphered from the above-mentioned studies, and the references were added to the end of the paragraphs. Kindly specify what exactly is needed from our end 

English errors should be corrected. For example: “it” in Line 232, “In-hospital” in Line 85, “The all cause in hospital” in Line 134, … Reply: unformally changed to In-hospital 

The space issue still persists in some places. For example: Line 171, Line 120, Line 225, Line 131, …There is a font and style issue in Lines 66-67.Some texts have different colors (gray). For example, Lines 179-192, 215-219.  Reply: changes made. 

Table captions, references, author affiliations, ... have incorrect styles and fonts! Reference 11 is just a hyperlink and does not show more details! Affiliation numbers are not superscripts. Reply: Affiliation numbers of authors are superscripts: We were not specified which style to use for references, normally those changes are made during the final proof read version from mdpi ( based on our previous experience), and we are given it again for rechecking 

Figure 4, which is revised, is out of the page borders! Reply: We cross checked: the image is within borders 

The limitations that I mentioned in the previous comments have not been added, and the limitations section is not a subsection of the discussion; instead, the authors have removed section numbers! Reply: limitations have been added: The limitations have been added as subsection of discussion 

Why are all captions and manuscript title now written in capital letters? Reply: The title and captions are not caps anymore 

What does ** mean? Reply: changed 

Line 74: Simirnov was corrected to Smirnov in v2, but now it is again incorrect in v3!
Line 57: O2HA0QZ or 02HA0QZ (zero or O letter)? Reply: It's Smirnov, correction made 

Regarding the extra free spaces, It is a technically issue arising from conversion from pages ( that we work on) to word. Normally spacing and orientation gets corrected by the minor english editing services at the end of peer review and mdpi sends us the manuscript to check and proof read. This coversion from pages to word does affect some of the spaces and orientation. Kindly understand this limitation 

Round 4

Reviewer 1 Report

Comments and Suggestions for Authors

The manuscript seems to revised in many aspects. There are still some issues that need to be addressed. The abstract has a lot of errors and needs to be fully revised. Although more relevant research is now added, it seems that all reference numbers in the text do not match with the references list (see comment 14).

  1. Line 14: Why outcomes and hospital start with capital letters?
  2. Why "Trends" in line 16 starts with a capital letter?
  3. Line 16: "2018-" finish the sentence.
  4. Line 17: fix the error "Table Hospital ..."
  5. Line 19: finish the sentence "Table .. "
  6. The authors should avoid just mentioning figure and table numbers in the results of the abstract and should instead summarize the findings in this section of abstract.
  7. There is still space issue, for example in line 20, line 115 (all- cause), line 125 (2.21 %), line 230 (30- Day should be 30-day), ...
  8. Line 20 fix english error "as well ..."
  9. Describe the meaning of abbreviations when they first appear, for example, VA ECMO in Line 69.
  10. Line 140 write fugure number (figure 6) and add "." at the end of sentences.
  11. Make sure that all references are correctly written. For example, in line 152 should be [11] or [12]? It seems that ref 11 is about intermacs!
  12. Line 199 correct "CVOID" to "COVID".
  13. Correct the limitations section in lines 218-220. "However, since ...".
  14. It seems that all reference numbers are incorrect. Line 213: says [20] but the MOMENTUM trial is [19] in ref list! Or in line 74, ICD code is written [11] but in the ref list it is [10]. All reference numbers should be checked.
Comments on the Quality of English Language

Line 14: Why outcomes and hospital start with capital letters?

Why "Trends" in line 16 starts with a capital letters?

Line 16: "2018-" finish the sentence.

Line 17: fix the error "Table Hospital ..."

Line 19: finish the sentence "Table .. "

The authors should avoid just mentioning figure and table numbers in the results of the abstract and should instead summarize the findings in this section of abstract.

There is still space issue, for example in line 20, line 115 (all- cause), line 125 (2.21 %), line 230 (30- Day should be 30-day), ...

Line 20 fix english error "as well ..."

Describe the meaning of abbreviations when they first appear, for example, VA ECMO in Line 69.

Line 140 write fugure number (figure 6) and add "." at the end of sentences.

Author Response

  1. Line 14: Why outcomes and hospital start with capital letters?
  2. Why "Trends" in line 16 starts with a capital letter?
  3. Line 16: "2018-" finish the sentence.
  4. Line 17: fix the error "Table Hospital ..."
  5. Line 19: finish the sentence "Table .. "
  6. The authors should avoid just mentioning figure and table numbers in the results of the abstract and should instead summarize the findings in this section of abstract.

REPLY: all the changes tend to be related to the abstract that we have completely revised 

  1. There is still space issue, for example in line 20, line 115 (all- cause), line 125 (2.21 %), line 230 (30- Day should be 30-day), ...
  2. Line 20 fix english error "as well ..."
  3. Describe the meaning of abbreviations when they first appear, for example, VA ECMO in Line 69.
  4. Line 140 write fugure number (figure 6) and add "." at the end of sentences.
  5. Line 199 correct "CVOID" to "COVID".

REPLY: changes made 

  1. Correct the limitations section in lines 218-220. "However, since ...".: REPLY: changes made 

  1. It seems that all reference numbers are incorrect. Line 213: says [20] but the MOMENTUM trial is [19] in ref list! Or in line 74, ICD code is written [11] but in the ref list it is [10]. All reference numbers should be checked. REPLY: momentum trial is 20 in the reference list, 19 being a study on line infections. The reference for the ICD codes has been corrected, with 11 being cited for the official website of the ICD.  reference 12 is the INTERMACS report that is quoted where we have referenced the INTERMACS study report from 2012 - 2022 

We have also gone through the manuscript once again and corrected grammatical errors and spelling mistakes. However, line spacing issues tend to come up due to changes in the layout we do in our side versus the  version that gets uploaded. Kindly excuse us.  

Round 5

Reviewer 1 Report

Comments and Suggestions for Authors

There are still space issues in the text particularly in the abstract. For example, 2016 - 2022 in line 16 should be 2016-2022, 2.21 % in line 22 should be 2.21%, 2018 - 2022 in line 33 should be 2018-2022. Line 17 should be In-hospital. 

Comments on the Quality of English Language

There are still space issues in the text particularly in the abstract. For example, 2016 - 2022 in line 16 should be 2016-2022, 2.21 % in line 22 should be 2.21%, 2018 - 2022 in line 33 should be 2018-2022. Line 17 should be In-hospital. 

Author Response

There are still space issues in the text particularly in the abstract. For example, 2016 - 2022 in line 16 should be 2016-2022, 2.21 % in line 22 should be 2.21%, 2018 - 2022 in line 33 should be 2018-2022. Line 17 should be In-hospital. 

REPLY: changes made